# Idiopathic and secondary restless legs syndrome during pregnancy in Japan: Prevalence, clinical features and delivery-related outcomes

Chikara Yoshimura[1,2,3]*, Hisatomi Arima[1], Hironobu Amagase[4], Mizuko Takewaka[4], Kazuko Nakashima[4], Chikako Imaoka[4], Nanami Miyanaga[4], Hirotsugu Obama[4], Masaki Fujita[2], Shin-ichi Ando[3]

1 Department of Preventive Medicine and Public Health, Fukuoka University, Fukuoka, Japan, 2 Faculty of Medicine, Department of Respiratory Medicine, Fukuoka University, Fukuoka, Japan, 3 Sleep Apnea Center, Kyushu University Hospital, Fukuoka, Japan, 4 Amagase Obstetrics and Gynecology Clinic, Onojo, Japan

* ychikara@fukuoka-u.ac.jp

**Data Availability Statement:** All relevant data are within the paper and its Supporting Information files.

## Abstract

### Objective

The aim of this study was to investigate prevalence of idiopathic and secondary restless legs syndrome (RLS) according to pregnancy trimester, and its effects on delivery-related outcomes among pregnant women in Japan.

### Methods

This was a single-center, prospective observational study. One hundred eighty-two consecutive pregnant women participated in the study from June 2014 to March 2016. Participants were interviewed and examined in the second and third trimesters of pregnancy and 1 month after delivery. At each term, RLS was identified by a research assistant and then specialist in sleep medicine based on the diagnostic criteria of the International Restless Legs Syndrome Study Group. Delivery-related data was collected from medical charts. RLS was classified as idiopathic RLS, which originally existed before the index pregnancy, or secondary RLS, which newly appeared during the index pregnancy.

### Results

The prevalence of RLS was 4.9% (idiopathic 3.3%, secondary 1.6%) in the second trimester, 5.0% (idiopathic 0.0%, secondary 5.0%) in the third trimester, and 0.6% (idiopathic 0.0%, secondary 0.6%) after delivery. Prolonged labor, emergency Cesarean section, and arrest of labor tended to be more frequent in idiopathic and/or second RLS (all p<0.05).

### Conclusions

The prevalence of RLS during pregnancy was 4–5% and decreases after delivery in current Japan. The presence of RLS was associated with an increase in some delivery-related

**Funding:** This study was supported by Japan Society for the Promotion of Science KAKENHI Grant number 15K08737 and The Fukuoka University Internal Research Competitive Funds (Grant No.197006). CY and MF have grants for my institution from Fukuda Denshi and Fukuda Lifetec Kyushu. SA has grants for Kyushu University Hospital from Teijin Pharma, Philips Respironics, Fuji Zerox, Daiichi Sankyo, Astellas Pharma. CY received lecture fees from Fukuda Denshi, Fukuda Lifetec Kyushu, Pacific Medico, Philips Respironics, Daiichi Sankyo, Takeda, Otsuka. HA received lecture fees from Bayer, Daiichi Sankyo, Fukuda Denshi, MSD, Takeda, Teijin Pharma and fees for consultancy from Kyowa Kirin. SA received lecture fees from Teijin Pharma. The funders had no role in study design, data collection and analysis, decision to publish, or preparation of the manuscript. The specific roles of these authors are articulated in the 'author contributions' section.

**Competing interests:** SA has grants for Kyushu University Hospital from Teijin Pharma, Philips Respironics, Fuji Zerox, Daiichi Sankyo, Astellas Pharma. CY received lecture fees from Fukuda Denshi, Fukuda Lifetec Kyushu, Pacific Medico, Philips Respironics, Daiichi Sankyo, Takeda, Otsuka. HA received lecture fees from Bayer, Daiichi Sankyo, Fukuda Denshi, MSD, Takeda, Teijin Pharma and fees for consultancy from Kyowa Kirin. SA received lecture fees from Teijin Pharma. There are no patents, products in development or marketed products to declare. The remaining authors have indicated no conflicts of interest. This does not alter our adherence to PLOS ONE policies on sharing data and materials.

**Abbreviations:** BMI, body mass index; BP, blood pressure; ESS, Epworth Sleepiness Scale; IRLS, International restless legs syndrome rating scale; RLS, restless legs syndrome; SD, standard deviations.

outcomes. Early detection and treatment of RLS during pregnancy may be beneficial to safe delivery for pregnant women.

## 1. Introduction

Sleep disorders such as difficulty in initiating sleep and/or in maintaining sleep due to midway awakenings are frequently observed during pregnancy [1] in Japan [2,3] as well as in other countries around the world. Increase in sleep disorders during pregnancy has been shown to be attributable to hormonal imbalance, increased abdominal pressure, elevated total blood volume etc [4,5].

Restless legs syndrome (RLS) is a sleep disorder [6,7], which is characterized by an unpleasant dysesthesia of the legs that begin after rest and is relieved with movement. The prevalence of RLS has been shown to increase up to 20% [6,8] during pregnancy according to previous reports from Europe, the United States and Japan. In current Japan, however, because of increased number of pregnant women who take folic acid [9] and/or iron agents, the prevalence of RLS during pregnancy might have decreased. In addition, some previous studies suggested that RLS during pregnancy is associated with poor delivery-related outcomes [7,10].

Current knowledge of the effect of pregnancy-related RLS is mainly derived from Western populations [11,12], and it is not clear to what extent these findings apply to a Japanese population. Furthermore, there have been limited studies that reported a prevalence of RLS according to type (idiopathic or secondary) or pregnancy period.

Our hypothesis was that prevalence of RLS during pregnancy in current Japan was lower than that in previous studies conducted in Europe, the United States and Japan. We also hypothesized that RLS during pregnancy was associated with the increased risks of delivery-related complications.

The aim of this study was to investigate the prevalence of idiopathic and secondary RLS according to pregnancy trimester, and it's effects on delivery-related outcomes among pregnant women in Japan.

## 2. Methods

### 2.1. Study design

This was a single-center, prospective observational study.

### 2.2. Study participants, inclusion criteria

From June 2014 to March 2016, pregnant women (aged 20–49 years, stable pregnancies, ability to undergo examination during the pregnancy) who visited the Amagase Obstetrics and Gynecology Clinic for regular examinations during the second trimester (gestational age from 14 to 27 weeks) were randomly invited to participate in the study. We excluded women with a history of depression or severe diseases such as heart failure, cancer or kidney disease. A total of 182 pregnant women (10.9% of 1,671 pregnant women who visited the clinic during the study period) who provided informed consent to the study were included in the present analysis. This study was approved by the Ethics Committee of Kyushu University Hospital and written informed consent was obtained from all participants before enrolment.

## 2.3. Data collection

Participants were interviewed and examined in the second and third trimesters of pregnancy and 1 month after delivery. At each visit, body height and weight were measured without shoes, and body mass index (BMI) was calculated. Blood pressure was measured twice, using a mercury sphygmomanometer with appropriately sized cuff on the right arm after 5-minute rest, and average values of the two measurements were used in the present analysis. Abdominal circumference in the standing position and fundal height in the supine position were also measured by trained staff using standard methods.

Information on pre-pregnancy status (BMI, parity, family history of RLS, alcohol consumption, smoking status, and histories of hypertension and diabetes) was also collected at the second trimester visit. History of alcohol intake was defined as pre-pregnancy habitual intake of ≥20g alcohol for once a week or more, and history of smoking was defined as habit of pre-pregnancy smoking. We defined pre-pregnancy hypertension and diabetes mellitus based on clinical diagnosis or through the use of blood pressure or glucose lowering treatment.

Delivery-related data (1. maternal data: gestational age [time from the last menstrual period to the delivery] at delivery, premature [before 37 weeks of gestation] or postmature delivery [after 42 weeks of gestation], delivery time [time from the onset of labor until delivery of the placenta], normal vaginal delivery [delivery without any abnormalities], induced labor [use of medication or other techniques to induce contractions], vacuum extraction [use of a vacuum device], prolonged labor [>30 hours for primiparas or >15 hours for multiparas], elective Cesarean section, emergency Cesarean section which was performed when the mother and child were judged to be in danger, premature rupture of the membrane [rupture of egg membrane before the start of parturition], arrest of labor [complete cessation of progress], and amount of bleeding; 2. fetal data: pH, $PaO_2$ and $PaCO_2$ based on umbilical-cord blood gas analysis, birth weight, gender, and Apgar score [at 1 minute and at 5 minutes]) were collected from medical charts.

## 2.4. Laboratory examination

Casual blood and urinary samples were collected at the second and third trimester visits and at 1 month after delivery. Full blood count was performed using an automated hemocytometer. Blood urea nitrogen was measured using the urease LED UV method; serum creatinine, using the enzymatic method; serum iron [13] and total/unsaturated iron-binding capacity, using the PSAP method; and serum ferritin, folic acid and vitamin B12, using the CLEIA method. Proteinuria and glycosuria were assessed using the dipstick method.

## 2.5. Diagnosis of RLS

At each visit, information on the presence of Allen's tetralogy (1. the urge to move the legs usually but not always accompanied by or felt to be caused by uncomfortable and unpleasant sensations in the legs; 2. the urge to move the legs and any accompanying unpleasant sensations which begin or worsen during periods of rest or inactivity such as lying down or sitting; 3. the urge to move the legs and any accompanying unpleasant sensations that are partially or totally relieved by movement, [such as walking or stretching] at least as long as the activity continues; 4. the urge to move the legs and any accompanying unpleasant sensations during rest or inactivity which only occur or are worse in the evening or night than during the day) [14] was obtained by trained staff and the final diagnosis was made by a specialist in sleep medicine based on the diagnostic criteria of the international restless legs syndrome study group [15].

RLS was classified as idiopathic RLS, which originally existed before the index pregnancy, or secondary RLS, which newly appeared during the index pregnancy [16]. Secondary RLS

was further divided into new-onset or persistent across 2 or more trimesters. Symptoms and severity of RLS were assessed using the International restless legs syndrome rating scale (IRLS) [17] and the Epworth Sleepiness Scale (ESS) [18]. Information on the use of medications which may improve symptoms of RLS (i.e., iron, folic acid, and multivitamins) was also collected from interviews. Presence, type, symptoms, and severity of pre-pregnancy RLS was also assessed at the second trimester visit.

## 2.6. Statistical analysis

Data was presented as means (standard deviations [SD]) or n (%). Delivery-related data was compared using the Kruskal-Wallis test or chi-square test, as appropriate. $P < 0.05$ was considered statistically significant. JMP version 9.0.0 was used for statistical analysis.

## 3. Results

Pre-pregnancy characteristics of participants are shown in Table 1. Average age was 31.9 years and average BMI was 22.7 kg/m$^2$. Frequency of multipara was 47.3%, family history of RLS 0.5% and pre-pregnancy alcohol intake 26.4%.

Table 2 shows clinical features at each study visit. From the second to third trimester, BMI increased from 22.7 to 24.4 kg/m$^2$, abdominal circumference from 84.3 to 92.5 cm, fundal height from 19.9 to 28.7 cm, and UIBC from 63.3 to 72.2 μmol/L. On the other hand, Vitamin B12 decreased from 163.2 to 133.0 pmol/L. Frequency of proteinuria increased from 15.9% to 25.3%. At one month after delivery, BMI decreased to 22.0 kg/m$^2$ and frequency of proteinuria decreased to 8.5%.

Fig 1 shows prevalence of idiopathic and secondary RLS. Prevalence of RLS was 4.9% (idiopathic 4.9%) before pregnancy, 4.9% (idiopathic 3.3%, secondary new 1.6%) in the second trimester, and 5.0% (secondary new-onset 4.4%, secondary persistence 0.6%) in the third trimester. Prevalence of RLS after delivery was 0.6% (secondary persistence 0.6%). Details of clinical course and use of the medications in all cases of restless legs syndrome were shown in S1 Fig. Table 3 shows symptoms, severity and supplementary medication that can affect symptoms according to presence and type of RLS. The IRLS scores for idiopathic and secondary RLS ranged from 15 to 20 throughout the study period. ESS scores were 5.0–5.8 before pregnancy, 9.4–10.6 in the second trimester, 9.8–9.9 in the third trimester and 9.0–10.6 after delivery in idiopathic or secondary RLS. There were no significant differences in ESS scores across the 3 groups of idiopathic RLS, secondary RLS and no RLS. In the third trimester of pregnancy, 60–78% of the participants had supplementary iron, but there were no significant

**Table 1. Pre-pregnancy characteristics.**

|  | Pre-pregnancy (n = 182) |
|---|---|
| Age(years) | 31.9±4.2 |
| Body mass index (kg/m$^2$) | 20.9±2.8 |
| Multipara, n (%) | 86(47.3%) |
| Family history of restless legs syndrome symptoms, n (%) | 1(0.5%) |
| Alcohol intake, n (%) | 48(26.4%) |
| Smoking, n (%) | 4(2.2) |
| Hypertension, n (%) | 1(0.5%) |
| Diabetes, n (%) | 0(0.0%) |
| Endometriosis, n (%) | 3(1.6%) |

Values are means (SD) or N (%).

**Table 2. Clinical features at each study visit.**

|  | Second trimester (n = 182) | Third trimester (n = 160) | After delivery (n = 159) |
|---|---|---|---|
| Body mass index (kg/m$^2$) | 22.7±2.5 | 24.4±2.6 | 22.0±2.5 |
| Systolic BP (mmHg) | 108.1±11.0 | 111.2±10.6 | 111.4±11.6 |
| Diastolic BP (mmHg) | 58.3±7.9 | 62.1±7.9 | 64.0±9.2 |
| Abdominal circumference (cm) | 84.3±6.2 | 92.5±6.1 | – |
| Fundal height (cm) | 19.9±2.1 | 28.7±2.4 | – |
| White blood cells (×10$^9$/L) | 8.7±1.9 | 8.0±1.9 | – |
| Red blood cells (×10$^{12}$/L) | 3.7±0.3 | 3.8±0.3 | – |
| Hemoglobin (g/L) | 112±9 | 113±8 | – |
| Hematocrit (L/L) | 0.34±0.03 | 0.35±0.02 | – |
| Platelets (×10$^9$/L) | 244±50 | 231±49 | – |
| Blood urea nitrogen (BUN) (mmol/L) | 2.8±0.6 | 2.8±0.8 | – |
| Creatinine (μmol/L) | 39.8±5.3 | 44.2±6.2 | – |
| Iron (μmol/L) | 13.9±6.8 | 15.0±10.1 | – |
| Total iron binding capacity (TIBC) (μmol/L) | 77.2±11.5 | 87.2±10.5 | – |
| Unsaturated iron binding capacity (UIBC) (μmol/L) | 63.3±15.5 | 72.2±16.0 | – |
| Ferritin (pmol/L) | 24.5±19.6 | 23.1±18.7 | – |
| Folate (nmol/L) | 22.5±13.4 | 19.9±12.2 | – |
| Vitamin B12 (pmol/L) | 163.2±59.2 | 133.0±51.8 | – |
| Proteinuria, n (%) | 29(15.9%) | 43(25.3%) | 14(8.5%) |
| Glycosuria, n (%) | 20(11.0%) | 23(13.5%) | 1(0.6%) |

Values are means (SD) or N (%).

BP: Blood pressure.

differences across the 3 groups. Gestational diabetes and endometriosis appeared in one and three women, while there were no cases of pregnancy induced hypertension or preeclampsia. There were no significant associations of maternal age, body mass index, hypertension, gestational diabetes [19], preeclampsia, and endometriosis with perinatal RLS (S1 Table).

Table 4 shows delivery-related data according to presence and type of restless legs syndrome. There were no significant differences in gestational age across the 3 groups, but prolonged labor, emergency Cesarean section, and arrest of labor tended to be more frequent in idiopathic and/or second RLS (all p<0.05). There were no significant differences in fetal factors across the 3 groups of idiopathic RLS, secondary RLS and no RLS (all p>0.3).

## 4. Discussion

To our knowledge, this is the first study in current Japan to report the prevalence of idiopathic and secondary RLS according to pregnancy trimester. The prevalence of RLS was 4.9% before pregnancy, 5.0% in the second trimester, 5.0% in the third trimester, and 0.6% after delivery. Although prevalence rates of RLS remained constant at 5% during pregnancy, new-onset cases of secondary RLS were observed in both the second and third trimesters. It was also suggested that RLS during pregnancy is associated with prolonged labor, emergency Cesarean section and/or arrest of labor.

In prior epidemiological studies, diagnosis of RLS has been made based on questionnaires, telephone surveys and/or interviews. After change of the diagnostic criteria in 2010 [15], in Europe, prevalence rates of RLS during pregnancy based on questionnaires have been reported to be 18–27% in the second trimester and 30–38% in the third trimester [12] while those based

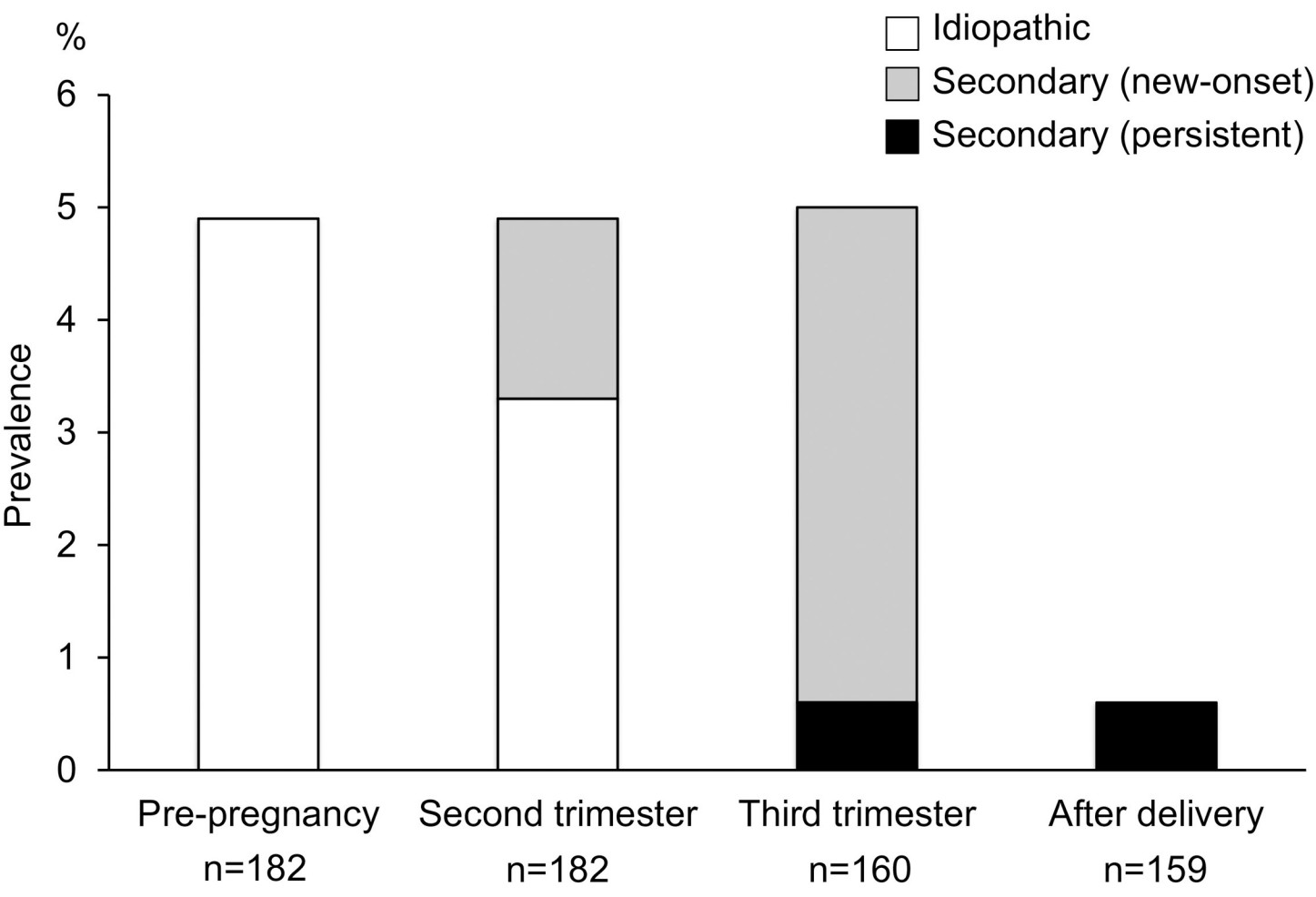

**Fig 1. Prevalence of idiopathic and secondary restless legs syndrome.**

on interviews have been reported to be 11% and 12%. In Asia, prevalence rates based on questionnaires have been reported to be 17% in the second trimester and 21% in the third trimester [20] while those based on interviews have been reported to be 11% and 18% [21,22]. In Japan, however, there have been no epidemiological studies that reported prevalence rates of RLS during pregnancy using the new diagnostic criteria. The present analysis used the new diagnostic criteria and demonstrated prevalence rates of RLS during pregnancy based on interviews in current Japan, which tended to be lower than those in prior studies in Asia, as well as in other regions of the world. Discrepancies in results may be attributable to differences in study design, settings (one question) [2], participants (family history [10]), medications (iron, folic acid and multivitamins, which has been shown to increase dopamine in the brain and, subsequently, to decrease the symptoms of RLS) [23,24] etc.

In the present analysis, prevalence of idiopathic RLS decreased from 3.3% in the 2nd trimester to 0% in the 3rd trimester. There has still be limited publications which reported trend in prevalence of RLS by type (idiopathic vs secondary) from 2nd to 3rd trimester. An observational study of 1584 pregnant women in China reported higher prevalence of idiopathic RLS in 3rd trimester (5.2%) than in 2nd trimester (4.3%) while this was a cross-sectional study and did not follow participants from 2nd to 3rd trimester [21]. Another longitudinal study of 642 pregnant women in Italy reported comparable prevalence rates of idiopathic RLS between 2nd and 3rd

**Table 3. Symptoms, severity and supplementary medication which can affect symptoms according to presence and type of restless legs syndrome.**

| | Pre-Pregnancy (n = 182) | Second trimester (n = 182) | Third trimester (n = 160) | After delivery (n = 159) |
|---|---|---|---|---|
| IRLS | | | | |
| Idiopathic RLS | 15.1±2.5 | 19.4±6.9 | – | – |
| Secondary RLS | – | 18.3±10.8 | 15.8±5.9 | 20.0* |
| No RLS | – | – | – | – |
| Epworth sleepiness scale | | | | |
| Idiopathic RLS | 5.8±3.1 | 9.4±4.3 | 9.8±5.9 | 10.6±5.6 |
| Secondary RLS | 5.0±3.3 | 10.6±4.1 | 9.9±3.6 | 9.0±3.4 |
| No RLS | 6.1±3.9 | 8.7±4.7 | 8.7±4.4 | 7.7±4.8 |
| Supplementary Medication | | | | |
| Iron | | | | |
| Idiopathic RLS | 0(0.0%) | 4(44.4%) | 7(77.8%) | 1(11.1%) |
| Secondary RLS | 0(0.0%) | 2(20.0%) | 6(60.0%) | 2(20.0%) |
| No RLS | 9(5.5%) | 49(30.0%) | 95(62.9%) | 57(38.8%) |
| Folate | | | | |
| Idiopathic RLS | 0(0.0%) | 7(77.8%) | 3(33.3%) | 2(22.2%) |
| Secondary RLS | 2(20.0%) | 4(40.0%) | 3(30.0%) | 1(10.0%) |
| No RLS | 22(13.5%) | 96(58.9%) | 51(33.8%) | 26(17.7%) |
| Multivitamins | | | | |
| Idiopathic RLS | 1(11.1%) | 2(22.2%) | 1(11.1%) | 1(11.1%) |
| Secondary RLS | 2(20.0%) | 1(10.0%) | 1(10.0%) | 0(0.0%) |
| No RLS | 6(3.7%) | 18(11.0%) | 14(8.6%) | 2(1.4%) |

Values are means (SD) or N (%).

IRLS: International restless legs syndrome rating scale, RLS: Restless legs syndrome.

*SD is not calculable.

trimesters but approximately 11% of participants with idiopathic RLS in the 2nd trimester showed improvement in RLS symptoms in the 3rd trimester [24]. The reasons for various time courses in prevalence of idiopathic RLS in 2nd and 3rd trimesters are not clear but discrepancies might be attributable to differences in study design (longitudinal or cross-sectional), ethnicity [21,25,26], use of medications (iron, folic acid and multivitamins) [26], susceptibility to medications [23,24] etc.

In the present analysis, new-onset cases of secondary RLS were observed in both the second and third trimesters. Prior epidemiological studies also suggested that new RLS cases were observed in the third trimester as well as in the second [25,27–29]. Findings of the present analysis and prior epidemiological studies support the importance of multiple exams for the screening of RLS during the pregnancy period.

Delivery-related outcomes in secondary RLS have been reported from prior epidemiological studies. In some studies, secondary RLS did not affect the frequency of Cesarean section, fetal body weight, gestational age or Apgar scores at 1- or 5-minute [30]. On the other hand, it has also been reported that secondary RLS may affect gestational age and fetal body weight [31]. In this study, RLS did not clearly affect fetal status although it may be associated with prolonged labor, emergency Cesarean section and/or arrest of labor. Early resolution of RLS could reduce the cesarean section rate and the risk of complications such as bladder injury [32]. Based on the findings of this study and previous reports, RLS during pregnancy may affect delivery and, possibly, fetal status.

**Table 4. Delivery-related data according to presence and type of restless legs syndrome.**

|  | Idiopathic RLS (n = 9) | Secondary RLS (n = 10) | No RLS (n = 141) | p value |
|---|---|---|---|---|
| Gestational age, weeks | 40.0±1.1 | 39.2±1.2 | 39.0±1.3 | 0.081 |
| Gestational age, days | 282.3±6.5 | 277.9±9.2 | 275.8±9.3 | 0.091 |
| Premature delivery, n (%) | 0 (0.0%) | 0 (0.0%) | 5 (3.3%) | 0.723 |
| Postmature delivery, n (%) | 0 (0.0%) | 0 (0.0%) | 0 (0.0%) | 0.999 |
| Delivery time, min | 586.4±586.3 | 516.2±756.4 | 480.2±452.4 | 0.807 |
| NVD, n (%) | 2 (22.2%) | 5 (50.0%) | 83 (55.7%) | 0.144 |
| Induced labor, n (%) | 0 (0.0%) | 2 (20.0%) | 14 (9.5%) | 0.329 |
| Vacuum extraction, n (%) | 1 (11.1%) | 1 (10.0%) | 11 (7.4%) | 0.886 |
| Prolonged labor, n (%) | 1 (11.1%) | 1 (10.0%) | 1(0.7%) | 0.010 |
| Elective CS, n (%) | 1 (11.1%) | 2 (20.0%) | 22 (14.9%) | 0.857 |
| Emergency CS, n (%) | 2 (22.2%) | 0 (0.0%) | 6 (4.1%) | 0.035 |
| PROM, n (%) | 1 (11.1%) | 0 (0.0%) | 11 (7.4%) | 0.608 |
| Arrest of labor, n (%) | 1 (11.1%) | 0 (0.0%) | 1 (0.7%) | 0.018 |
| Bleeding, ml | 420.1±224.7 | 429.2±225.0 | 394.1±276.7 | 0.677 |
| pH of umbilical-code blood | 7.31±0.13 | 7.31±0.04 | 7.32±0.06 | 0.300 |
| $PaO_2$ of umbilical-code blood | 19.6±4.7 | 20.6±5.4 | 20.2±6.0 | 0.960 |
| $PaCO_2$ of umbilical-code blood | 50.6±13.0 | 49.6±5.0 | 49.2±8.6 | 0.820 |
| Birth weight, g | 3216±452 | 3040±351 | 3066±399 | 0.718 |
| Boys, n (%) | 5 (55.6%) | 5 (60.0%) | 69 (45.7%) | 0.593 |
| Apgar score |  |  |  |  |
| 1-minute score | 9.0±0.0 | 9.0±0.0 | 8.9±0.6 | 0.507 |
| 5-minute score | 9.8±0.4 | 9.8±0.4 | 9.7±0.5 | 0.862 |

Values are means (SD) or N (%).

RLS: Restless legs syndrome, NVD: Normal vaginal delivery, CS: Cesarean section, PROM: Premature rupture of the membrane.

There are several limitations to this study. First, because this is a single-clinic study and recruitment rate was 10.9% of all pregnant women who visited the clinic, sample size is somewhat small and findings of this study may be affected by selection bias and referral bias. Second, a small number of participants moved to other clinics/hospitals due to relocation or complications and did not come back to follow-up assessments. Third, prevalence of RLS may have been underestimated because of frequent use of medications that may improve symptoms of RLS (i.e., iron, folic acid, and multivitamins). Fourth, prevalence of RLS may have been overestimated because we did not collect information on RLS mimics (e.g., leg cramps, positional discomfort, myalgia, venous stasis, leg edema and arthritis). Fifth, we made the diagnosis of "idiopathic RLS before pregnancy" in the 2nd trimester of pregnancy, accuracy may be somewhat limited.

In conclusion, prevalence of RLS during pregnancy was 4.9% in the second trimester and 5.0% in the third trimester in current Japan. New-onset RLS cases secondary to pregnancy were observed both in the second and the third trimesters of pregnancy. Furthermore, presence of RLS was associated with increases in some delivery-related outcomes. Early detection and treatment of RLS using multiple screening interviews during pregnancy may be beneficial to safe delivery and childbirth as well as the quality of life of pregnant women.

## Supporting information

**S1 Fig. Clinical course and use of the medications in all cases of restless legs syndrome.**
(PDF)

**S1 Table. Relationship of maternal age, body mass index, hypertension, gestational diabetes mellitus and endometriosis with Perinatal RLS.**
(PDF)

## Acknowledgments

We would like to thank Dr. Yoshihiko Amagase, honorary director, Amagase Obstetrics and Gynecology Clinic, Onojo, for conducting accurate and reliable case registration and data management for this clinical study.

## Author Contributions

**Conceptualization:** Chikara Yoshimura, Shin-ichi Ando.

**Data curation:** Chikara Yoshimura, Hironobu Amagase, Mizuko Takewaka, Kazuko Nakashima, Chikako Imaoka, Nanami Miyanaga, Hirotsugu Obama.

**Formal analysis:** Chikara Yoshimura, Hisatomi Arima, Shin-ichi Ando.

**Funding acquisition:** Chikara Yoshimura, Shin-ichi Ando.

**Investigation:** Chikara Yoshimura, Hironobu Amagase, Mizuko Takewaka, Kazuko Nakashima, Chikako Imaoka, Nanami Miyanaga, Hirotsugu Obama.

**Methodology:** Chikara Yoshimura, Shin-ichi Ando.

**Project administration:** Chikara Yoshimura, Shin-ichi Ando.

**Resources:** Chikara Yoshimura, Shin-ichi Ando.

**Software:** Chikara Yoshimura, Hironobu Amagase.

**Supervision:** Hisatomi Arima, Masaki Fujita, Shin-ichi Ando.

**Validation:** Chikara Yoshimura, Hironobu Amagase, Mizuko Takewaka, Kazuko Nakashima, Chikako Imaoka, Nanami Miyanaga, Hirotsugu Obama, Shin-ichi Ando.

**Visualization:** Chikara Yoshimura, Hisatomi Arima, Shin-ichi Ando.

**Writing – original draft:** Chikara Yoshimura, Hisatomi Arima, Shin-ichi Ando.

**Writing – review & editing:** Hironobu Amagase, Mizuko Takewaka, Kazuko Nakashima, Chikako Imaoka, Nanami Miyanaga, Hirotsugu Obama, Masaki Fujita.

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
