## [Decision Letter · Decision Letter 0]

18 Aug 2020

PONE-D-20-18639

Idiopathic and secondary restless legs syndrome during pregnancy in Japan: prevalence, clinical features and delivery-related outcomes

PLOS ONE

Dear Dr. Yoshimura,

Thank you for submitting your manuscript to PLOS ONE. After careful consideration, we feel that it has merit but does not fully meet PLOS ONE’s publication criteria as it currently stands. Therefore, we invite you to submit a revised version of the manuscript that addresses the points raised during the review process.

We look forward to receiving your revised manuscript.

Kind regards,

Antonio Simone Laganà, M.D., Ph.D.

Academic Editor

PLOS ONE

Additional Editor Comments:

The topic of the manuscript is interesting. Nevertheless, the reviewers raised several concerns: considering this point, I invite authors to perform the required major revisions.

Journal Requirements:

"This study was supported by Japan Society for the Promotion of Science KAKENHI Grant number 15K08737 and The Fukuoka University Internal Research Competitive Funds (Grant No.197006).  CY and MF have grants for my institution from Fukuda Denshi and Fukuda Lifetec Kyushu.  SA has grants for Kyushu University Hospital from Teijin Pharma, Philips Respironics, Fuji Zerox, Daiichi Sankyo, Astellas Pharma.  CY received lecture fees from Fukuda Denshi, Fukuda Lifetec Kyushu, Pacific Medico, Philips Respironics, Daiichi Sankyo, Takeda, Otsuka.  HA received lecture fees from Bayer, Daiichi Sankyo, Fukuda Denshi, MSD, Takeda, Teijin Pharma and fees for consultancy from Kyowa Kirin.  SA received lecture fees from Teijin Pharma.  The other of authors report no conflicts of interest.

We note that you received funding from commercial sources: Fukuda Denshi, Fukuda Lifetec Kyushu, Teijin Pharma, Philips Respironics, Fuji Zerox, Daiichi Sankyo, Astellas Pharma.

Please provide an amended Competing Interests Statement that explicitly states these commercial funders, along with any other relevant declarations relating to employment, consultancy, patents, products in development, marketed products, etc.

Reviewers' comments:

Reviewer's Responses to Questions

**Comments to the Author**

1. Is the manuscript technically sound, and do the data support the conclusions?

Reviewer #1: Partly

Reviewer #2: Yes

Reviewer #3: Partly

2. Has the statistical analysis been performed appropriately and rigorously? 

Reviewer #1: Yes

Reviewer #2: Yes

Reviewer #3: No

3. Have the authors made all data underlying the findings in their manuscript fully available?

Reviewer #1: Yes

Reviewer #2: Yes

Reviewer #3: No

4. Is the manuscript presented in an intelligible fashion and written in standard English?

Reviewer #1: Yes

Reviewer #2: Yes

Reviewer #3: Yes

5. Review Comments to the Author

Reviewer #1: • This is a single center study titled ‘Idiopathic and secondary restless legs syndrome during pregnancy in Japan: prevalence, clinical features and delivery-related outcomes’

• The research was a prospective observational study. One hundred eighty-three consecutive pregnant women participated in the study from June 2014 to March 2016.

• Participants were interviewed and examined in the second and third trimesters of pregnancy and 1 month after delivery.

• RLS diagnosis was made on the diagnostic criteria of the International Restless Legs Syndrome Study Group (2010).

My opinions and comments are as follows:

Introduction:

1- Literature information on the subject was not given sufficiently

2- The hypothesis is not revealed clearly

Methods:

From June 2014 to March 2016, consecutive pregnant women (age ≥18 years) who visited the Amagase Obstetrics and Gynecology Clinic during the second trimester were invited to participate in the study. A total of 183 pregnant women who provided informed consent to the study were included.

1- The number of patients is insufficient for a prevalence study

2- From June 2014 to March 2016, how many consecutive pregnant women (age ≥18 years) applied to the Amagase Obstetrics and Gynecology Clinic and how many accepted to participate in the study?

3- International Classification fo Sleep Disorders was updated in 2014 as ICSD-3, and RLS diagnostic criteria was also updated as follows;

RLS Diagnostic Criteria:

Criteria A-C must be met

A. An urge to move the legs, usually accompanied by or thought to be caused by uncomfortable and unpleasant sensations in the legs.1,2 These symptoms must:

1. Begin or worsen during periods of rest or inactivity such as lying down or sitting;

2. Be partially or totally relieved by movement, such as walking or

stretching, at least as long as the activity continues; and

3. Occur exclusively or predominantly in the evening or night rather

than during the day.

B. The above features are not solely accounted for as symptoms of another medical or a behavioral condition (e.g., leg cramps, positional discomfort, myalgia, venous stasis, leg edema, arthritis, habitual foot tapping).

C. The symptoms of RLS cause concern, distress, sleep disturbance, or

impairment in mental, physical, social, occupational, educational,

behavioral, or other important areas of functioning.

4- The authors should have used these criteria. At least leg cramps, positional discomfort, myalgia, venous stasis, leg edema, arthritis should have been questioned.

Results and discussion:

1- Before pregnancy idiopathic RLS prevalence was 4.9%, but in the second trimester it decreased to 2.7%

RLS, which existed before pregnancy in some pregnant women, appears to have improved in pregnancy

Also secondary persistence was 0.6% in the second trimester

RLS, which existed in the second trimester in some pregnant women, appears to have improved in the third trimester

These rates have not been examined separately

2- Secondary persistance in the third trimester and postpartum seems the same (0.6%), this situation has not been discussed either

3- New diagnosed diseases during pregnancy; like gestational DM and preeclampsia was not questioned

4- Relation between maternal age, weight and additional diseases with RLS have not been investigated

Reviewer #2: The authors should be applauded for their work describing RLS during pregnancy in Japan. This is an important topic that is often overlooked, possibly because it falls between two specialties (sleep medicine/neurology and obstetrics).

I have several questions regarding the manuscript as well as some thoughts on how it may be improved.

Major concerns:

I am confused as to the trajectory of RLS symptoms in this cohort. Based on the results section as well as table 3 and Figure one it appears that the following is true:

1) The prevalence of RLS in young women in Japan is around 5%. This seems plausible.

2) During the 2nd trimester the prevalence actually drops. While some new cases arise these are more than offset by the apparent resolution of some of the idiopathic cases.

3) By the 3rd trimester everyone who had RLS before pregnancy no longer has RLS, ie the RLS has resolved. Others have developed RLS during the pregnancy which keeps the prevalence around 5%

4) Following delivery almost all of the RLS resolves. The prevalence is now very low and no one who had RLS prior to pregnancy had a return of symptoms.

In most cases pregnancy exacerbates pre-existing RLS. In fact pre-pregnancy RLS is the strongest predictor of pregnancy related RLS (https://doi.org/10.1016/j.sleep.2008.06.011) However, these finding suggest that pregnancy improves RLS symptoms to the same degree that it causes it. This finding could be due to several different possibilities. A) I am misinterpreting the data, in which case further explanation would be appreciated B) RLS did improve for many of the subjects, which if is the case the authors should elaborate on this in detail including reasons for why they think this could be the case. C) There is a methodological problem such as not asking about RLS symptoms at subsequent visits or having an imprecise diagnostic tool.

Minor concerns:

1) The wording of the methods section of abstract gives the impression that a sleep medicine specialist evaluated each subject during each visit. The wording of the methods section itself gives the impression that a research assistant asked a series of questions based IRLSSG / ICSD-3 criteria and then a sleep medicine specialist reviewed the results at a later time. This should clarified.

2) It would be beneficial to briefly discuss the role of iron in the RLS as the authors mention that being on iron supplements may have affected their results.

3) For the Suzuki 2003 paper the estimated prevalence of RLS during pregnancy in Japan was based off on a single question and not the four criteria that the authors of this study used. The authors mention this but should elaborate further as it could help to explain the much higher prevalence that the Suzuki 2003 paper reports.

4) Was there a certain time of day that the labs were drawn? Serum iron levels have strong circadian fluctuations. (https://doi.org/10.1016/j.clinbiochem.2010.08.023)

5) Other papers which may add to the manuscript. https://doi.org/10.1016/j.sleep.2009.04.005

https://doi.org/10.1016/j.genhosppsych.2009.11.016

https://doi.org/10.5664/jcsm.3704

https://doi.org/10.1016/j.smrv.2014.10.009

Reviewer #3: I was pleased to revise the manuscript entitled “Idiopathic and secondary restless legs syndrome during pregnancy in Japan: prevalence, clinical features and delivery-related outcomes” (Manuscript Number: PONE-D-20-18639).

This study was approved by the Ethics Committee of Kyushu University Hospital and written informed consent was obtained from all participants before enrolment.

In general, this manuscript was aimed to investigate prevalence of idiopathic and secondary restless legs syndrome (RLS) according to pregnancy trimester, and its effects on delivery-related outcomes among pregnant women in Japan. In my honest opinion, the topic is interesting enough to attract the readers’ attention. Methodology is accurate and conclusions are supported by the data analysis. Nevertheless, authors should clarify some points.

In general, the Manuscript may benefit from some major revisions, as suggested below:

- All the text needs a language revision in order to improve some typos and grammatical errors.

- I would suggest checking the guidelines for the Authors to conform the manuscript. In example, the results section needs to be reported before the methods.

- Methods. I would suggest providing more details about the recruitment of patients. Which was the use modality? Which proportion of the actual total population referring to the hospital was included? Which is the response rate? Are characteristics of non-responders available or at least reason to non-participate? Which information was provided to the patients for recruitment? These pieces of information are paramount to identify possible biases.

- At which gestational age was the recruitment allowed?

- When the idiopathic RLS was assessed? How was it actually defined? Before pregnancy or before a certain gestational age?

- Which was the frequency of obstetrics visits? Was it the same for all patients?

- Discussion. Possible biases, such as non-responder bias, attrition bias, and referral bias need to be discussed. The association between cesarean section and RLS could be related to the fact that patients with obstetric pathologies underwent an higher number of obstetrics evaluation with higher chance of RLS diagnosis.

- How would the authors interpret the disappearing of idiopathic RLS? Was it assessed only once during pregnancy? It is unclear how a condition present before pregnancy is no more reported later in pregnancy.

- The observed association between RLS and cesarean section is interesting, and I would suggest, at least briefly, stressing more complications related to cesarean section, such as bladder injuries (refer to: PMID: 30877907), to stress the importance to reduce cesarean section rate.

- Did the authors include in the analysis possible other background pathologies, such as endometriosis, or gestational diabetes? In this regard, I would suggest discussing about other possible obstetrics complications that could be investigated in terms of association with RLS. Refer to: PMID: 31903997; DOI: 10.1007/s10397-015-0901-9; PMID: 32046116.

6. PLOS authors have the option to publish the peer review history of their article (what does this mean?). If published, this will include your full peer review and any attached files.

Reviewer #1: No

Reviewer #2: No

Reviewer #3: No

---

## [Author Response · Author response to Decision Letter 0]

17 Jan 2021

Response to Reviewer #1 

Thank you for your useful suggestions.

First of all, we would like to report that age range and the number of participants were not reported accurately in the previous manuscript. The participants were not aged ≥18 yeas but were aged 20-49 years, and the number of participants was not 183 but was 182. However, the change did not affect the results of the paper. According to this correction, we have amended Abstract, Results, Tables and Figures. We sincerely apologize for the error in the previous manuscript.

We have attempted to address your suggestions as follows:

Comment - Introduction 1: Literature information on the subject was not given sufficiently

Response: Thank you very much for your useful suggestions. We have now added more detailed literature information of RLS in the 2nd paragraph of the Introduction section (lines 8-16, page 4).

“Restless legs syndrome (RLS) is a sleep disorder, which is characterized by an unpleasant and itchy dysesthesia of the legs that begin after rest and is relieved with movement during pregnancy. The prevalence of RLS has been shown to increase up to 20% during pregnancy according to previous reports from Europe, the United States and Japan. In current Japan, however, because of increased number of pregnant women who take folic acid and/or iron agents, the prevalence of RLS during pregnancy might have decreased. In addition, some previous studies suggested that RLS during pregnancy is associated with poor delivery-related outcomes.”

Comment - Introduction 2: The hypothesis is not revealed clearly

Response: Thank you very much for your useful suggestion. We have added our hypotheses in the 4th paragraph of the section as follows (from line 2, page 5 to line 5, page 5). 

“Our hypothesis was that prevalence of RLS during pregnancy in current Japan was lower than that in previous studies conducted in Europe, the United States and Japan. We also hypothesized that RLS during pregnancy was associated with the increased risks of delivery-related complications.”

Comment - Methods 1: The number of patients is insufficient for a prevalence study

Response: Our main hypothesis was that prevalence of RLS during pregnancy in current Japan was lower than that in previous studies conducted in Europe (in 2010’s), the United States (in 2010’s) and of Japan (in early 2000’s) (approximately 20%). In this study, the prevalence of RLS was 4.9 % (95% CI 4.86-4.94%) in the 2nd and 5.0% (95% CI 4.96-5.04%) in the 3rd trimester, that were significantly lower than 20% as reported in previous studies. When power calculation was conducted our study had more than 99% power to test our main hypothesis. We are sorry, but we had determined that this number of cases is sufficient.

Comment - Methods 2: From June 2014 to March 2016, how many consecutive pregnant women (age ≥18 years) applied to the Amagase Obstetrics and Gynecology Clinic and how many accepted to participate in the study?

Response: The medical staff of the clinic randomly invited pregnant women to participate in this study. Finally, a total of 182 pregnant women (10.9% of 1671 pregnant women who visited clinic during the study period) participated in this study. We have clarified this fact in 2.2. Study Participants, Inclusion Criteria section of the Methods as follows (from line 16, page 5 to line 2, page 5).

“From June 2014 to March 2016, pregnant women (aged 20-49 years, stable pregnancies, ability to undergo examination during the pregnancy) who visited the Amagase Obstetrics and Gynecology Clinic for regular examinations during the second trimester were randomly invited to participate in the study. We excluded women with a history of depression or severe diseases such as heart failure, cancer or kidney disease. A total of 182 pregnant women (10.9% of 1,671 pregnant women who visited clinic during the study period) who provided informed consent to the study were included in the present analysis.”

Comment - Methods 3: International Classification for Sleep Disorders was updated in 2014 as ICSD-3, and RLS diagnostic criteria was also updated as follows;

RLS Diagnostic Criteria:

Criteria A-C must be met

A. An urge to move the legs, usually accompanied by or thought to be caused by uncomfortable and unpleasant sensations in the legs.1,2 These symptoms must:

1. Begin or worsen during periods of rest or inactivity such as lying down or sitting;

2. Be partially or totally relieved by movement, such as walking or

stretching, at least as long as the activity continues; and

3. Occur exclusively or predominantly in the evening or night rather

than during the day.

B. The above features are not solely accounted for as symptoms of another medical or a behavioral condition (e.g., leg cramps, positional discomfort, myalgia, venous stasis, leg edema, arthritis, habitual foot tapping).

C. The symptoms of RLS cause concern, distress, sleep disturbance, or

impairment in mental, physical, social, occupational, educational,

behavioral, or other important areas of functioning.

The authors should have used these criteria. At least leg cramps, positional discomfort, myalgia, venous stasis, leg edema, arthritis should have been questioned.

Response: Thank you very much for your comments. Because the protocol and data collection procedures were established before the announcement of the ICSD-3 criteria in 2014. Unfortunately, we did not collect information on RLS mimics including leg cramps, positional discomfort, myalgia, venous stasis, leg edema and arthritis by the ICSD-3 criteria. As a result, we might have overestimated the prevalence rates of RLS. We have added this limitation in the 5th paragraph of the Discussion section (lines 16-18, page 12) as follows.

“Fourth, prevalence of RLS may have been overestimated because we did not collect information on RLS mimics (e.g., leg cramps, positional discomfort, myalgia, venous stasis, leg edema and arthritis).”

Comment - Results and Discussion 1:

Before pregnancy idiopathic RLS prevalence was 4.9%, but in the second trimester it decreased to 2.7% RLS, which existed before pregnancy in some pregnant women, appears to have improved in pregnancy. Also secondary persistence was 0.6% in the second trimester RLS, which existed in the second trimester in some pregnant women, appears to have improved in the third trimester. These rates have not been examined separately.

Response: Thank you very much for your suggestion. In order to clarify detailed time course of each pregnant woman with idiopathic or secondary RLS, we have added supplementary Figure 1. Idiopathic RLS decreased from 4.9% before pregnancy to 3.3% at the 2nd trimester. This is because the Cases 7-9 of Idiopathic RLS in supplementary Figure 1 recovered from RLS symptoms after pregnancy. Secondary RLS decreased from 1.6% at the 2nd trimester to 0.6% at the 3rd trimester. This is because the Cases 2 and 3 of Secondary RLS in supplementary Figure 1 recovered from RLS symptoms at the 3rd trimester.

Comment - Results and Discussion 2: Secondary persistence in the third trimester and postpartum seems the same (0.6%), this situation has not been discussed either

Response: We are sorry to make you misunderstand results of this paper. The case of the persistent secondary RLS in the 3rd trimester (Case 1 of secondary RLS onset at 2nd trimester in supplementary Figure 1) is not the same person as the case of persistent secondary RLS after delivery (Case 7 of secondary RLS onset at 3rd trimester in supplementary Figure 1). In order to clarify detailed time course of each pregnant woman with idiopathic or secondary RLS, we have added supplementary Figure 1.

Comment - Results and Discussion 3: New diagnosed diseases during pregnancy; like gestational DM and preeclampsia was not questioned

Response: Thank you very much for pointing out a very important point. Gestational diabetes appeared in one woman, while there were no cases of preeclampsia or pregnancy induced hypertension. We have added this fact in the 3rd paragraph of the Results section as follows (from line 25, page 9 to line 2, page 10).

“Gestational diabetes and endometriosis appeared in one and three women, while there were no cases of pregnancy induced hypertension or preeclampsia.”

Comment - Results and Discussion 4: Relation between maternal age, weight, and additional diseases with RLS have not been investigated

Response: According to your suggestion, the relationship of maternal age, body mass index, hypertension, and gestational diabetes with perinatal RLS was shown in Supplementary Table1. There were no significant associations of maternal age, body mass index, hypertension, and gestational diabetes with perinatal RLS.

Supplementary Table1. Relationship of maternal age, body mass index, hypertension, gestational diabetes mellitus and endometriosis with Perinatal RLS 

Response to Reviewer #2 

Thank you for your useful suggestions.

First of all, we would like to report that age range and the number of participants were not reported accurately in the previous manuscript. The participants were not aged ≥18 yeas but were aged 20-49 years, and the number of participants was not 183 but was 182. However, the change did not affect the results of the paper. According to this correction, we have amended Abstract, Results, Tables and Figures. We sincerely apologize for the error in the previous manuscript.

We have attempted to address your suggestions as follows:

Major comment 1. I am confused as to the trajectory of RLS symptoms in this cohort. Based on the results section as well as table 3 and Figure one it appears that the following is true:

1) The prevalence of RLS in young women in Japan is around 5%. This seems plausible.

2) During the 2nd trimester the prevalence actually drops. While some new cases arise, these are more than offset by the apparent resolution of some of the idiopathic cases.

3) By the 3rd trimester everyone who had RLS before pregnancy no longer has RLS, ie the RLS has resolved. Others have developed RLS during the pregnancy which keeps the prevalence around 5%

4) Following delivery almost all the RLS resolves. The prevalence is now very low and no one who had RLS prior to pregnancy had a return of symptoms.

In most cases pregnancy exacerbates pre-existing RLS. In fact, pre-pregnancy RLS is the strongest predictor of pregnancy related RLS (https://doi.org/10.1016/j.sleep.2008.06.011) However, these finding suggest that pregnancy improves RLS symptoms to the same degree that it causes it. This finding could be due to several different possibilities. A) I am misinterpreting the data, in which case further explanation would be appreciated B) RLS did improve for many of the subjects, which it is the case the authors should elaborate on this in detail including reasons for why they think this could be the case. C) There is a methodological problem such as not asking about RLS symptoms at subsequent visits or having an imprecise diagnostic tool.

Response: We are sorry for your confusion. As you mentioned, pre-pregnancy RLS had been reported to be the most powerful predictor of pregnancy-related RLS, while many pregnant women with idiopathic RLS recovered during pregnancy in our study.

You raised three possibilities (A, B and C), but recovery of RLS symptoms during pregnancy is not your misinterpretation(A). It is not methodological problem (C), neither, because same procedures were used for diagnosis of RLS at each visit. Therefore, RLS did improve for many of the subjects in our study (B）.

There might be a few possible reasons for recovery of RLS symptoms during pregnancy. One possible reason is frequent use of iron, folic acid, and multivitamins, which have been shown to improve RLS symptoms, among pregnant women with idiopathic/secondary RLS in this study. Another possible reason is Japanese government recommends pregnant women to use folic acid in order to prevent fetal malformations.

With regard to recovery of RLS symptoms after delivery, pregnant women will not be deprived of iron by the foetation after childbirth. Improvement of iron stores after delivery has been shown to increase dopamine in the brain, and to improve RLS symptoms. 

Minor comment 1. The wording of the methods section of abstract gives the impression that a sleep medicine specialist evaluated each subject during each visit. The wording of the methods section itself gives the impression that a research assistant asked a series of questions based IRLSSG / ICSD-3 criteria and then a sleep medicine specialist reviewed the results at a later time. This should clarify.

Response: Thank you very much. A research assistant asked a series of questions based IRLSSG criteria and then a sleep medicine specialist reviewed the results at a later time. We changed Abstract as follow.

“At each term, RLS was identified by a research assistant and then specialist in sleep medicine based on the diagnostic criteria of the International Restless Legs Syndrome Study Group.”

Minor comment 2. It would be beneficial to briefly discuss the role of iron in the RLS as the authors mention that being on iron supplements may have affected their results.

Response: Thank you very much for your advice. Oral administration of iron raises iron and ferritin in the cerebrospinal fluid, eventually increases dopamine in the brain, and decreases the symptoms of RLS. We have added this discussion in the 2nd paragraph of the Discussion section as follows (line 11-14, page 11).

“Discrepancies in results may be attributable to differences in study design, settings, participants, medications (iron, folic acid and multivitamins, which has been shown to increase dopamine in the brain and, subsequently, to decrease the symptoms of RLS) etc.”

Minor comment 3. For the Suzuki 2003 paper the estimated prevalence of RLS during pregnancy in Japan was based off on a single question and not the four criteria that the authors of this study used. The authors mention this but should elaborate further as it could help to explain the much higher prevalence that the Suzuki 2003 paper reports.

Response: Thank you very much for your advice. Suzuki et al reported that the prevalence of RLS among pregnant women in Japan is highly estimated because of one question. We added the literature on the diagnostic criteria of RLS, and added that the reason why the diagnosis of RLS has changed due to the difference in the method is also one of the reasons why the result was different from Suzuki et al. We have added this discussion in the 2nd paragraph of the Discussion section as follows (line 12, page 11).

“Discrepancies in results may be attributable to differences in study design, settings (one question) Suzuki 2003, participants, medications (iron, folic acid and multivitamins, which has been shown to increase dopamine in the brain and, subsequently, to decrease the symptoms of RLS) etc.”

Minor comment 4. Was there a certain time of day that the labs were drawn? Serum iron levels have strong circadian fluctuations. 

(https://doi.org/10.1016/j.clinbiochem.2010.08.023)

Response: We are sorry, and added this paper (lines 14, page 7). Although serum iron has strong circadian fluctuations, it was difficult to extract only participants and collect blood at the same time. Therefore, we also measured serum ferritin, which has little fluctuation.

Minor comment 5. Other papers which may add to the manuscript. https://doi.org/10.1016/j.sleep.2009.04.005

https://doi.org/10.1016/j.genhosppsych.2009.11.016

https://doi.org/10.5664/jcsm.3704

https://doi.org/10.1016/j.smrv.2014.10.009

Response: Thank you very much for your advice. We have added the first and fourth literature in the References. Although this study didn’t have a family history of RLS, we have added the first literature in the 2nd paragraph of the Discussion section (lines 12, page 11), because the family history of RLS was very important.

We are sorry. The second literature was the result of an interview using the 2003’s IRLS criteria in South Korea, and since we were considering the result using the 2010’s IRLS criteria, we did not have adopted it.

We are sorry. We didn’t add the third literature, so we didn’t have analyzed the depression scale this time.

We have added the fourth literatures in the 2nd paragraph of the Introduction section (lines 11, page 4).

Response to Reviewer #3 

Thank you for your useful suggestions. 

First of all, we would like to report that age range and the number of participants were not reported accurately in the previous manuscript. The participants were not aged ≥18 yeas but were aged 20-49 years, and the number of participants was not 183 but was 182. However, the change did not affect the results of the paper. According to this correction, we have amended Abstract, Results, Tables and Figures. We sincerely apologize for the error in the previous manuscript.

We have attempted to address your suggestions as follows:

Major comment 1. All the text needs a language revision in order to improve some typos and grammatical errors.

Response: Thank you very much. We checked again with professional English editorial service for typos and grammatical errors.

Major comment 2. I would suggest checking the guidelines for the Authors to conform the manuscript. In example, the results section needs to be reported before the methods.

Response: Thank you very much for your advice. We have reviewed the guidelines for complying with the manuscript and changed the results section before the methods section.

Major comment 3. Methods. I would suggest providing more details about the recruitment of patients. Which was the use modality? Which proportion of the actual total population referring to the hospital was included? Which is the response rate? Are characteristics of non-responders available or at least reason to non-participate? Which information was provided to the patients for recruitment? These pieces of information are paramount to identify possible biases.

Response: According to your suggestion, we have added more details about the recruitment of patients in the Methods.

1. With regard to modality, pregnant women (aged 20-49 years, stable pregnancies, ability to undergo examination during the pregnancy) who visited the Amagase Obstetrics and Gynecology Clinic for regular examinations during the second trimester (gestational age from 14 to 27 weeks) were randomly invited to participate in the study. We excluded women with a history of depression or severe diseases such as heart failure, cancer, or kidney disease.

2. The proportion of 182 women included this study among the actual total population who visited the clinic during the study period (n=1,671) was 10.9%.

3. We are sorry but we did not record the number of invited women and are not able to calculate response rate.

4. Unfortunately, we do not have information on characteristics of 1489 pregnant women who did not participate in this study, because the ethics committee did not allow us to collect clinical information of women who did not provide consent to participate in this study. 

5. Participant information sheet for recruitment included objective, study procedures, risks, compensation, voluntary participation, confidentiality, cost, conflict of interest and study investigators. 

6. As described above, this is a single-clinic study and recruitment rate was 10.9% of all pregnant women who visited the clinic, sample size is somewhat small and findings of this study may be affected by selection bias. We have described this limitation in the 5th paragraph of the Discussion section (from line 9, page 12 to line 12, page 12). 

Major comment 4. At which gestational age was the recruitment allowed?

Response: Thank you very much for your advice. We recruited the pregnant women who were aged 20-49 years and gestational age from 14 weeks to 27 weeks in 2nd trimester.

Major comment 5. When the idiopathic RLS was assessed? How was it actually defined? Before pregnancy or before a certain gestational age?

Response: As we mentioned, the diagnosis of idiopathic RLS was not made before pregnancy, but it was diagnosed using information collected in the 2nd trimester of pregnancy. Therefore, accuracy in “idiopathic RLS before pregnancy” may be somewhat limited in study. We have added this limitation in the 5th paragraph of Discussion section (lines 19-20, page 12). 

“Fifth, we made the diagnosis of “idiopathic RLS before pregnancy” in the 2nd trimester of pregnancy, accuracy may be somewhat limited.”

Major comment 6. Which was the frequency of obstetrics visits? Was it the same for all patients?

Response: Thank you very much for your advice. Almost women visited every 2 weeks. However, there were some emergency visits such as anemia and constipation. Therefore, all pregnant women evaluated by regular visits.

Major comment 7. Discussion. Possible biases, such as non-responder bias, attrition bias, and referral bias need to be discussed. The association between cesarean section and RLS could be related to the fact that patients with obstetric pathologies underwent a higher number of obstetrics evaluation with higher chance of RLS diagnosis.

Response: Thank you very much for your advice. A research plan was made and implemented to reduce the bias as much as possible, but referral bias such as urgent admission to another hospital was unavoidable. I added it to the limitation. We have added this limitation in the 5th paragraph of Discussion section as follows (lines 10-12, page 12).

”sample size is somewhat small and findings of this study may be affected by selection bias and referral bias.”

Major comment 8. How would the authors interpret the disappearing of idiopathic RLS? Was it assessed only once during pregnancy? It is unclear how a condition present before pregnancy is no more reported later in pregnancy.

Response: Thank you very much for your advices. It was highly possible that idiopathic RLS was improved by oral administration of iron preparations due to anemia. Evaluations were made during the second and third trimester of pregnancy and after delivery. In order to clarify detailed time course of pregnant woman with idiopathic RLS, we have added supplementary Figure 1.

Major comment 9. The observed association between RLS and cesarean section is interesting, and I would suggest, at least briefly, stressing more complications related to cesarean section, such as bladder injuries (refer to: PMID: 30877907), to stress the importance to reduce cesarean section rate.

Response: Thank you very much for your comment. We added and quoted that early resolution of RLS could reduce the cesarean section rate and the risk of complications such as bladder injury. We have added this discussion in the 4th paragraph of Discussion section as follows (from line 4, page 12 to line 6, page 12).

“Early resolution of RLS could reduce the cesarean section rate and the risk of complications such as bladder injury.”

Major comment 10. Did the authors include in the analysis possible other background pathologies, such as endometriosis, or gestational diabetes? In this regard, I would suggest discussing about other possible obstetrics complications that could be investigated in terms of association with RLS. Refer to: PMID: 31903997; DOI: 10.1007/s10397-015-0901-9; PMID: 32046116.

Response: According to your suggestion, the relationship of maternal age, body mass index, possible obstetrics complications (hypertension, gestational diabetes, preeclampsia and endometriosis) with perinatal RLS was shown in Supplementary Table1. There were no significant associations of maternal age, body mass index, hypertension, gestational diabetes, preeclampsia, and endometriosis with perinatal RLS. We have added this discussion in the 3rd paragraph of Results section as follows (from line 25, page 9 to line 4, page 10).

“Gestational diabetes and endometriosis appeared in one and three women, while there were no cases of pregnancy induced hypertension or preeclampsia. There were no significant associations of maternal age, body mass index, hypertension, gestational diabetes, preeclampsia, and endometriosis with perinatal RLS (Supplementary Table1).”

---

## [Decision Letter · Decision Letter 1]

8 Mar 2021

PONE-D-20-18639R1

Idiopathic and secondary restless legs syndrome during pregnancy in Japan: prevalence, clinical features and delivery-related outcomes

PLOS ONE

Dear Dr. Yoshimura,

Thank you for submitting your manuscript to PLOS ONE. After careful consideration, we feel that it has merit but does not fully meet PLOS ONE’s publication criteria as it currently stands. Therefore, we invite you to submit a revised version of the manuscript that addresses the points raised during the review process.

We look forward to receiving your revised manuscript.

Kind regards,

Antonio Simone Laganà, M.D., Ph.D.

Academic Editor

PLOS ONE

Journal Requirements:

Additional Editor Comments (if provided):

Authors improved significantly the quality of the paper, following the recommendations of the Reviewers.

Nevertheless, one of them still has some concerns: for this reason, I invite authors to perform additional minor revisions.

Reviewers' comments:

Reviewer's Responses to Questions

**Comments to the Author**

1. If the authors have adequately addressed your comments raised in a previous round of review and you feel that this manuscript is now acceptable for publication, you may indicate that here to bypass the “Comments to the Author” section, enter your conflict of interest statement in the “Confidential to Editor” section, and submit your "Accept" recommendation.

Reviewer #2: (No Response)

Reviewer #3: All comments have been addressed

2. Is the manuscript technically sound, and do the data support the conclusions?

Reviewer #2: Partly

Reviewer #3: Yes

3. Has the statistical analysis been performed appropriately and rigorously? 

Reviewer #2: I Don't Know

Reviewer #3: Yes

4. Have the authors made all data underlying the findings in their manuscript fully available?

Reviewer #2: Yes

Reviewer #3: Yes

5. Is the manuscript presented in an intelligible fashion and written in standard English?

Reviewer #2: Yes

Reviewer #3: Yes

6. Review Comments to the Author

Reviewer #2: Thank you for the revisions.

In response to my primary question about the study, the authors state that "RLS did improve for many of the subjects in our study." However that under-emphasizes this finding and perhaps misses the point I was trying to make. While some studies have reported improvement in RLS symptoms during pregnancy for a minority of subjects with pre-existing RLS (11% on systemic review doi:10.1186/s12883-020-01709-0), to my knowledge none have reported complete resolution of RLS in such large number of cases.

Not only did RLS improve for some of the subjects in this study, 100% of the subjects with RLS prior to pregnancy had resolution of RLS by the third trimester. This fact is still not adequately discussed by the authors, particularly as it varies so much from established literature.

The statement re: the role of iron in RLS is simplistic and lacking a citation. The thought that use of iron supplements may account for some of the improvement in symptoms is reasonable, though that this may explain the large difference between the prevalence of this study and others is unconvincing as most countries have similar recommendations.

The description of RLS is also somewhat odd "Restless legs syndrome (RLS) is a sleep disorder(6) (7), which is characterized by an unpleasant and itchy dysesthesia of the legs that begin after rest and is relieved with movement during pregnancy." Itchiness is not necessarily present and the wording sounds as if RLS is specific to pregnancy.

Reviewer #3: I was pleased to revise the manuscript entitled “Idiopathic and secondary restless legs syndrome during pregnancy in Japan: prevalence, clinical features and delivery-related outcomes” (Manuscript Number: PONE-D-20-18639).

This study was approved by the Ethics Committee of Kyushu University Hospital and written informed consent was obtained from all participants before enrolment.

In general, this manuscript was aimed to investigate prevalence of idiopathic and secondary restless legs syndrome (RLS) according to pregnancy trimester, and its effects on delivery-related outcomes among pregnant women in Japan. In my honest opinion, the topic is interesting enough to attract the readers’ attention. Methodology is accurate and conclusions are supported by the data analysis. Moreover, the authors addressed all the suggested revisions, and I appreciated the manuscript improvement.

7. PLOS authors have the option to publish the peer review history of their article (what does this mean?). If published, this will include your full peer review and any attached files.

Reviewer #2: No

Reviewer #3: No

---

## [Author Response · Author response to Decision Letter 1]

21 Apr 2021

Response to Reviewer #2 

Thank you for your useful suggestions.

We have attempted to address your suggestions as follows:

Major comment 1. In response to my primary question about the study, the authors state that "RLS did improve for many of the subjects in our study." However that under-emphasizes this finding and perhaps misses the point I was trying to make. While some studies have reported improvement in RLS symptoms during pregnancy for a minority of subjects with pre-existing RLS (11% on systemic review doi:10.1186/s12883-020-01709-0), to my knowledge none have reported complete resolution of RLS in such large number of cases.

Response:

As you pointed out, our findings were different from those from previous publications. While, in our study, idiopathic RLS in the 2nd trimester disappeared in the 3rd trimester, a Chinese cross-sectional study reported higher prevalence of idiopathic RLS in the 3rd trimester than in the 2nd trimester. Another longitudinal study conducted in Italy reported comparable prevalence rates but 11% of pregnant women with idiopathic RLS reported improvement of symptoms. The reasons for various time courses in prevalence of idiopathic RLS in 2nd and 3rd trimesters are not clear but discrepancies might be attributable to differences in study design (longitudinal or cross-sectional), ethnicity, use of medications (iron, folic acid and multivitamins), susceptibility to medications etc. We have discussed this point in the 3rd paragraph of the Discussion (line 3-17, page 12) as follows.

“In the present analysis, prevalence of idiopathic RLS decreased from 3.3% in the 2nd trimester to 0% in the 3rd trimester. There has still be limited publications which reported trend in prevalence of RLS by type (idiopathic vs secondary) from 2nd to 3rd trimester. An observational study of 1584 pregnant women in China reported higher prevalence of idiopathic RLS in 3rd trimester (5.2%) than in 2nd trimester (4.3%) while this was a cross-sectional study and did not follow participants from 2nd to 3rd trimester. Another longitudinal study of 642 pregnant women in Italy reported comparable prevalence rates of idiopathic RLS between 2nd and 3rd trimesters but approximately 11% of participants with idiopathic RLS in the 2nd trimester showed improvement in RLS symptoms in the 3rd trimester. The reasons for various time courses in prevalence of idiopathic RLS in 2nd and 3rd trimesters are not clear but discrepancies might be attributable to differences in study design (longitudinal or cross-sectional), ethnicity, use of medications (iron, folic acid and multivitamins), susceptibility to medications etc.”

Major comment 2. Not only did RLS improve for some of the subjects in this study, 100% of the subjects with RLS prior to pregnancy had resolution of RLS by the third trimester. This fact is still not adequately discussed by the authors, particularly as it varies so much from established literature.

Response:

We sincerely apologize for inadequate discussion regarding significant reduction in prevalence rates of idiopathic RLS from 2nd to 3rd trimester. We have added discussion regarding this point in the 3rd paragraph of the Discussion (line 3-17, page 12) as follows.

“In the present analysis, prevalence of idiopathic RLS decreased from 3.3% in the 2nd trimester to 0% in the 3rd trimester. There has still be limited publications which reported trend in prevalence of RLS by type (idiopathic vs secondary) from 2nd to 3rd trimester. An observational study of 1584 pregnant women in China reported higher prevalence of idiopathic RLS in 3rd trimester (5.2%) than in 2nd trimester (4.3%) while this was a cross-sectional study and did not follow participants from 2nd to 3rd trimester. Another longitudinal study of 642 pregnant women in Italy reported comparable prevalence rates of idiopathic RLS between 2nd and 3rd trimesters but approximately 11% of participants with idiopathic RLS in the 2nd trimester showed improvement in RLS symptoms in the 3rd trimester. The reasons for various time courses in prevalence of idiopathic RLS in 2nd and 3rd trimesters are not clear but discrepancies might be attributable to differences in study design (longitudinal or cross-sectional), ethnicity, use of medications (iron, folic acid and multivitamins), susceptibility to medications etc.”

Major comment 3. The statement re: the role of iron in RLS is simplistic and lacking a citation. The thought that use of iron supplements may account for some of the improvement in symptoms is reasonable, though that this may explain the large difference between the prevalence of this study and others is unconvincing as most countries have similar recommendations.

Response: 

We apologize for lack of citation regarding lack of information regarding role of iron. We have now added reference 23 in the last sentence of the 2nd paragraph of the Discussion (line 1, page 12).

We agree with you that iron supplements cannot explain large differences in prevalence of idiopathic RLS between our study and previous studies. At the moment, the reasons for various time courses in prevalence of idiopathic RLS in 2nd and 3rd trimesters are not clear but discrepancies might be attributable to differences in study design (longitudinal or cross-sectional), ethnicity, use of medications (iron, folic acid and multivitamins), susceptibility to medications etc. We have discussed this point in the 3rd paragraph of the Discussion (line 3-17, page 12) as follows.

“In the present analysis, prevalence of idiopathic RLS decreased from 3.3% in the 2nd trimester to 0% in the 3rd trimester. There has still be limited publications which reported trend in prevalence of RLS by type (idiopathic vs secondary) from 2nd to 3rd trimester. An observational study of 1584 pregnant women in China reported higher prevalence of idiopathic RLS in 3rd trimester (5.2%) than in 2nd trimester (4.3%) while this was a cross-sectional study and did not follow participants from 2nd to 3rd trimester. Another longitudinal study of 642 pregnant women in Italy reported comparable prevalence rates of idiopathic RLS between 2nd and 3rd trimesters but approximately 11% of participants with idiopathic RLS in the 2nd trimester showed improvement in RLS symptoms in the 3rd trimester. The reasons for various time courses in prevalence of idiopathic RLS in 2nd and 3rd trimesters are not clear but discrepancies might be attributable to differences in study design (longitudinal or cross-sectional), ethnicity, use of medications (iron, folic acid and multivitamins), susceptibility to medications etc.”

---

## [Editor Report · Decision Letter 2]

26 Apr 2021

Idiopathic and secondary restless legs syndrome during pregnancy in Japan: prevalence, clinical features and delivery-related outcomes

PONE-D-20-18639R2

Dear Dr. Yoshimura,

We’re pleased to inform you that your manuscript has been judged scientifically suitable for publication and will be formally accepted for publication once it meets all outstanding technical requirements.

Kind regards,

Antonio Simone Laganà, M.D., Ph.D.

Academic Editor

PLOS ONE

Additional Editor Comments (optional):

Authors performed the required corrections. I am pleased to accept this paper for publication.
---

## [Editor Report · Acceptance letter]

29 Apr 2021

PONE-D-20-18639R2 

Idiopathic and secondary restless legs syndrome during pregnancy in Japan: prevalence, clinical features and delivery-related outcomes 

Dear Dr. Yoshimura:

I'm pleased to inform you that your manuscript has been deemed suitable for publication in PLOS ONE. Congratulations! Your manuscript is now with our production department. 

Kind regards, 

on behalf of

Dr. Antonio Simone Laganà 

Academic Editor

PLOS ONE